# Hyperbaric Oxygen Therapy Alleviates Paclitaxel-Induced Peripheral Neuropathy Involving Suppressing TLR4-MyD88-NF-κB Signaling Pathway

**DOI:** 10.3390/ijms24065379

**Published:** 2023-03-11

**Authors:** Shih-Hung Wang, Shu-Hung Huang, Meng-Chien Hsieh, I-Cheng Lu, Ping-Ruey Chou, Ming-Hong Tai, Sheng-Hua Wu

**Affiliations:** 1School of Medicine, College of Medicine, Kaohsiung Medical University, Kaohsiung 807, Taiwan; 2Division of Plastic Surgery, Department of Surgery, Kaohsiung Medical University Hospital, Kaohsiung 807, Taiwan; 3Department of Surgery, School of Medicine, College of Medicine, Kaohsiung Medical University, Kaohsiung 807, Taiwan; 4Regeneration Medicine and Cell Therapy Research Center, Kaohsiung Medical University, Kaohsiung 807, Taiwan; 5Graduate Institute of Medicine, College of Medicine, Kaohsiung Medical University, Kaohsiung 807, Taiwan; 6Division of Plastic Surgery, Department of Surgery, Kaohsiung Municipal Siaogang Hospital, Kaohsiung 812, Taiwan; 7Division of Plastic Surgery, Department of Surgery, Kaohsiung Municipal Ta-Tung Hospital, Kaohsiung 801, Taiwan; 8Department of Anesthesiology, Kaohsiung Municipal Siaogang Hospital, Kaohsiung 812, Taiwan; 9Department of Anesthesiology, Kaohsiung Medical University Hospital, Kaohsiung 807, Taiwan; 10Department of Anesthesiology, School of Medicine, College of Medicine, Kaohsiung Medical University, Kaohsiung 807, Taiwan; 11Institute of Biomedical Sciences, National Sun Yat-Sen University, Kaohsiung 804, Taiwan; 12Department of Anesthesiology, Kaohsiung Municipal Ta-Tung Hospital, Kaohsiung 801, Taiwan

**Keywords:** chemotherapy-induced peripheral neuropathy, Paclitaxel, hyperbaric oxygen therapy, Toll-like receptor 4, myeloid differentiation factor 88 (MyD88)

## Abstract

Paclitaxel (PAC) results in long-term chemotherapy-induced peripheral neuropathy (CIPN). The coexpression of transient receptor potential vanilloid 1 (TRPV1) and Toll-like receptor 4 (TLR4) in the nervous system plays an essential role in mediating CIPN. In this study, we used a TLR4 agonist (lipopolysaccharide, LPS) and a TLR4 antagonist (TAK-242) in the CIPN rat model to investigate the role of TLR4-MyD88 signaling in the antinociceptive effects of hyper-baric oxygen therapy (HBOT). All rats, except a control group, received PAC to induce CIPN. Aside from the PAC group, four residual groups were treated with either LPS or TAK-242, and two of them received an additional one-week HBOT (PAC/LPS/HBOT and PAC/TAK-242/HBOT group). Mechanical allodynia and thermal hyperalgesia were then assessed. The expressions of TRPV1, TLR4 and its downstream signaling molecule, MyD88, were investigated. The mechanical and thermal tests revealed that HBOT and TAK-242 alleviated behavioral signs of CIPN. Immunofluorescence in the spinal cord dorsal horn and dorsal root ganglion revealed that TLR4 overexpression in PAC- and PAC/LPS-treated rats was significantly downregulated after HBOT and TAK-242. Additionally, Western blots showed a significant reduction in TLR4, TRPV1, MyD88 and NF-κB. Therefore, we suggest that HBOT may alleviate CIPN by modulating the TLR4-MyD88-NF-κB pathway.

## 1. Introduction

Paclitaxel (PAC) is a microtubule-stabilizing chemotherapeutic agent actively used for a broad spectrum of cancers [1]. However, up to 80% of cancer patients develop chemotherapy-induced peripheral neuropathy (CIPN) during or after PAC chemotherapy [2]. PAC-induced peripheral neuropathy is characterized by long-term and dose-dependent neuropathic pain symptoms, including tingling, burning, pain, and numbness, which not only reduce patients’ quality of life but also affect their motor function in severe cases [3]. Currently, pharmacological and nonpharmacological treatments for ameliorating CIPN have been discussed at length but are still equivocal and limited [4,5,6]. Therefore, preventive and therapeutic strategies for CIPN are urgently required.

Diverse causes of neuropathic pain are associated with excessive inflammation [7]. The pathomechanisms of PAC-induced CIPN include changes in myelinated axons, mitochondrial dysfunction, neuroinflammation and mechanical hypersensitivity mediated through ion channels, such as transient receptor potential vanilloid type 1 (TRPV1) [8]. TRPV1, a non-selective cation channel with high calcium (Ca^2+^) permeability, is primarily found in the spinal cord, dorsal root ganglion (DRG) and other nociceptive primary sensory neurons [9,10]. Recently, TRPV1 has been proven to play a key role in PAC-induced neuropathic pain symptoms [11,12,13,14,15]. Upregulated TRPV1 expression was noted in the DRG and the spinal cord of PAC-treated rats [12], whereas TRPV1 antagonists markedly reduced pain hypersensitivity. However, TRPV1 expression is not upregulated directly by PAC, but through Toll-like receptor 4 (TLR4) downstream signaling [13], a crucial role for the neuroinflammatory response [16]. A previous study also indicated that TLR4 signaling in the DRG and spinal cord plays a role in the initiation and maintenance of PAC-induced CIPN.

TLR4 recruits myeloid differentiation factor 88 (MyD88) to activate nuclear factor kappa B (NF-κB), promoting the release of inflammatory cytokines and the increased sensitivity of TRPV1 [13]. In DRG neurons, TRPV1 and TLR4 receptors are coexpressed, and the increase in TRPV1-expressing DRG neurons after PAC treatment is dependent on the activation of TLR4 [13]. Moreover, treatment with a TLR4 antagonist prevented CIPN development during chemotherapy in rats [13,17,18]. In the spinal cord dorsal horn, TLR4 is expressed predominantly in glial cells and causes the release of cytokines and chemokines to potentiate the function of the presynaptic TRPV1 receptor [12]. Our previous study demonstrated that PAC increases TLR4 and TRPV1 expression in both the spinal cord and DRG [19]. In brief, PAC-induced acute and persistent pain are highly attributed to the neuroinflammatory activation of TLR4/TRPV1 axis signaling [12,20].

Hyperbaric oxygen therapy (HBOT), which involves the administration of a high oxygen concentration (100%) at environmental pressure greater than 1 atmosphere absolute (ATA) [21], is clinically applied for a wide range of medical conditions, including non-healing ischemic wounds, post-radiation injuries, decompression sickness, burns, carbon monoxide intoxication and diabetic ulcers [22]. In addition, it is effective for chronic neuropathic pain treatment [23,24,25]. Despite the increased oxidative stress in HBOT, more and more studies have indicated that the right protocol of HBOT can also cause an elevation in antioxidants, which maintain the balance of the oxidative stress state [26]. Our previous research revealed that simultaneously receiving HBOT during PAC treatment resulted in early and persistent suppression of CIPN and produced more favorable reversal effects. Furthermore, HBOT decreased PAC-induced TLR4, TRPV1 and microglia activation in both the spinal cord dorsal horn and DRG [19].

Although the mechanisms underlying the development of PAC-induced CIPN remain undefined, previous studies indicated that TLR4 plays a key role. This study explores the anti-neuropathic potential of HBOT in PAC-induced CIPN with TLR4 agonist (lipopolysaccharide, LPS) or antagonist (TAK-242). The alleviation of CIPN was evaluated through pain behavior tests and the expression of myelin basic protein (MBP) in the sciatic nerve. TLR4 and its downstream signaling molecules MyD88 and NF-κB were assessed to further validate the neuroprotective and anti-inflammatory effects of HBOT on the CIPN rat model.

## 2. Results

### 2.1. Mechanical and Thermal Withdrawal Threshold

Behavior tests were assessed every two days before the rats were sacrificed (Figure 1A). This section provides the results of the mechanical allodynia (Figure 1B) and thermal hyperalgesia (Figure 1C) assessment. In the PAC/LPS/HBOT group, rats that received a 1-week period of HBOT had significant improvement in mechanical allodynia and thermal hyperalgesia evoked by PAC plus LPS treatment on D14 (*p* < 0.001, *p* < 0.001, respectively). When additional TAK-242 and/or HBOT was administered with PAC in the PAC/TAK-242/HBOT and PAC/TAK-242 groups, the deterioration of mechanical allodynia and thermal hyperalgesia was suppressed on D14 (all *p* < 0.001 compared with the PAC group). In the thermal behavior test, rats in the PAC/LPS/HBOT, PAC/TAK-242 and PAC/TAK-242/HBOT groups showed no obvious difference between the control rats on D14. However, on D14, the mechanical threshold in these three groups was still lower than the control group (*p* < 0.05). The improvement in thermal hyperalgesia after HBOT was better than the mechanical behavior test results, which was consistent with our earlier investigation [19].

### 2.2. Immunofluorescence (IF) Staining of MBP in Sciatic Nerve

We used myelin basic protein (MBP) to further investigate the ameliorative effect of HBOT on PAC-induced neuropathy. MBP, a mature myelin marker, was used to observe the morphological change in myelin in the sciatic nerve, and neurofilament 200 (NF200) was used to represent a marker of large-diameter myelinated nerve fibers through IF staining (Figure 2). Significantly decreased MBP expression was observed in the PAC and PAC/LPS groups (*p* < 0.05). After HBOT and TAK-242 treatment, the demyelination was significantly reduced in the PAC/LPS/HBOT, PAC/TAK-242 and PAC/TAK-242/HBOT groups. This result was consistent with the PAC-induced demyelination already reported [27].

### 2.3. IF Staining of TLR4/TRPV1 in the Lumbar Spinal Cord Dorsal Horn

In this study, the IF staining of TLR4 and TRPV1 expression (Figure 3) was evaluated in the L3–5 segments of the spinal cord dorsal horn. TRPV1 expression was majorly localized in the superficial layer, consistent with the study by Kamata et al. [28]. Compared with the control group, the PAC/LPS group had higher expression of TLR4 and TRPV1 in the superficial dorsal horn of the spinal cord (*p* < 0.01). TLR4 and TRPV1 expression showed no difference between the PAC and PAC/LPS groups. HBOT significantly suppressed the PAC plus LPS-induced overexpression of TLR4 and TRPV1 (*p* < 0.01). The PAC/TAK-242 and PAC/TAK-242/HBOT groups also had lower expression of TLR4, compared with the PAC groups (*p* < 0.05, *p* < 0.01, respectively). No significant difference in the expression TLR4 and TRPV1 was observed between the PAC/LPS/HBOT, PAC/TAK-242 and PAC/TAK-242/HBOT groups.

### 2.4. Western Blot Analysis of TLR4/MyD88/NF-κB/TRPV1 Expression of the Lumbar Spinal Cord Dorsal Horn

Similar to the results of IF staining in the lumbar spinal cord dorsal horn, Western blot analysis (Figure 4) revealed that the expression of TLR4 and TRPV1 was significantly upregulated in the PAC and PAC/LPS groups compared with that in the control group. Moreover, it revealed the markedly elevated levels of inflammatory proteins MyD88 and pNF-κB in the two groups (*p* < 0.01, compared to the control group). By contrast, HBOT and TAK-242 interventions significantly downregulated the expression of TLR4, TRPV1, MyD88 and pNF-κB that was induced by PAC or PAC/LPS. The difference in the expression of MyD88 and NF-κB was non-significant between the PAC/LPS/HBOT, PAC/TAK-242 and PAC/TAK-242/HBOT groups.

### 2.5. IF Staining of TLR4/MyD88 and TRPV1 in DRG

To obtain representative micrographs of TLR4 expression, IF staining of TLR4 (indicated by red) and DRG neurons (NeuN, indicated by green) was performed separately, and the images were merged (Figure 5). The signaling of the NeuN protein was positive both in the nucleus and cytoplasm of DRG neurons, which was consistent with the results of Anderson et al. [29]. IF staining revealed that TLR4 signaling was upregulated in the DRGs of the PAC/LPS group compared with the control group. TLR4 expression was significantly lower in the PAC/LPS/HBOT, PAC/TAK-242 and PAC/TAK-242/HBOT groups than in the PAC/LPS group (*p* < 0.001). No difference was observed in TLR4 expression of between the PAC/LPS/HBOT, PAC/TAK-242 and PAC/TAK-242/HBOT groups. The IF staining of TRPV1 (Figure 6) yielded similar results of upregulated expression of TRPV1 in the PAC/LPS group and significantly downregulated expression after TAK-242 and HBOT.

## 3. Discussion

Our previous study demonstrated that HBOT can alleviate PAC-induced neuropathy, potentially through downregulating TLR4-TRPV1 signaling [19]. In this study, LPS and TAK-242 were administrated as the TLR4 agonist and TLR4 antagonist, respectively, for investigating the anti-inflammatory effect of HBOT via modulation of TLR4/MyD88/NF-κB and TRPV1 pathway. Mechanical and thermal behavior tests revealed that the addition of LPS deteriorated PAC-induced neuropathy symptoms. IF staining of MBP, a mature Schwann cell marker, revealed that HBOT and TAK-242 improved sciatic nerve regeneration in PAC-treated rats. Most importantly, we found that the HBOT intervention can significantly ameliorate the CIPN symptoms enhanced by the TLR4 agonist and downregulate TLR4 signaling and TRPV1 expression. HBOT alone had no superior efficacy to TAK-242 alone and the TAK-242/HBOT group. Studies have discussed the relation between PAC and TLR-4 in LPS- or TAK-242-treated rat models [13,30,31]. Our study is the first to investigate the therapeutic potential and mechanism of HBOT for PAC-induced neuropathy by using a TLR4 agonist and TLR4 antagonist intervention.

The TLR4-mediated pathways are mainly divided into MyD88-dependent and non-MyD88-dependent pathways [32]. The involvement of the MyD88-dependent pathway in the PAC-induced neuropathy rat model has been reported [18]. MyD88 activates the NF-κB and mitogen-activated protein kinase pathways, causing the production of inflammatory cytokines, including interleukin-1β (IL-1β) and tumor necrosis factor-α (TNF-α) [33]. Li et al. [18] also reported that the TLR4 and MyD88 signaling pathways could be a potential therapeutic target in PAC-related CIPN. Our findings revealed that TLR4 and MyD88 expression significantly increased in the PAC and PAC/LPS groups compared with the control group. In addition to increased TLR4 and MyD88, we found increased NF-κB expression in PAC- and PAC/LPS-treated rats. HBOT significantly reduced the overexpression of TLR4, MyD88 and NF-κB, which suggests that the HBOT-mediated suppression of CIPN may be associated with the downregulation of TLR4, MyD88 and NF-κB signaling.

TLR4 is activated by the binding of LPS to the MD2 protein [34,35]. With a similar structure to LPS, PAC binds to the MD2 protein to activate the TLR4 receptor, causing CIPN [34,36,37]. In a mechanical behavior test by Ha et al. [37], the reduced threshold in LPS-injected rats lasted 3 days, while the reduced threshold in PAC-injected rats remained for more than 21 days. Similar to these findings, our results showed that the threshold of mechanical allodynia in the PAC/LPS group was lower than that in the PAC alone group till D10, and became no different thereafter; this result supports the suggestion that LPS-induced mechanical allodynia is a transient phenomenon. The thermal behavior test results revealed that the PAC/LPS group had a persistently lower thermal threshold than the PAC group, which is congruent with the finding of Ghefreh et al. that LPS-induced thermal hyperalgesia can persist over 14 days [38].

The TLR4 antagonist TAK-242 modulates TLR4 downstream signaling by binding to the Cys747 in the intracellular TLR4 domain [39,40]. Previous studies have shown that TAK-242 improves immune dysfunction and prevents endotoxemia-induced muscle wasting in vivo [41,42]. Li et al. and Zhang et al. have demonstrated the efficacy of TAK-242 for preventing PAC-induced neuropathy [13,43]. Consistent with their findings, our study revealed that HBOT produces similar effects to TAK-242 by attenuating the inflammatory response and mitigating neuropathic pain in behavioral tests, as well as decreasing the overexpression of TLR4 and TRPV1 in the spinal cord dorsal horn and DRG. More importantly, our findings showed no synergistic effect of TAK-242 and HBOT, which implied the possibility of their overlap in mechanism of action and supported the modulation of TLR4 as a major mechanism of HBOT in suppressing CIPN. HBOT has an effect on several molecular and cellular pathways that are important for neuronal recovery [44]. Liu et al. indicated that HBOT clinically alleviates hearing loss through the downregulation of TLR4, NF-κB and TNF-α in the peripheral blood of a subject [45]. HBOT has been shown to inhibit neuroinflammation through the regulation of LPS-induced NF-κB/MAPKs signaling pathways in astrocytes in vitro [46]. Figure 7 shows a summary of our findings, indicating that HBOT and TAK-242 can suppress the overexpression of TLR4, MyD88, NF-κB and TRPV1 in the spinal cord dorsal horn in the PAC-related CIPN models. We also suggest an overlapping mechanism between HBOT and TAK-242 due to the non-significant differences in various results.

There are several limitations in our study. Firstly, the effectiveness of HBOT for CIPN in the peripheral neuron system should be further investigated. Studies have shown significant and sustained epidermal nerve fiber loss in the CIPN mice model and patients with CIPN [47,48]. Cook et al. [49] demonstrated severely reduced axon area density, increased myelin abnormalities and a higher number of Schwann cell nuclei in the sciatic nerves of PAC-treated mice. Secondly, in contrast to previous studies that mentioned that TLR4 expression was solely noted in the superficial layer (lamina I-II) [13,28,50,51], our IF staining showed that TLR4 expression was not predominantly located in the superficial dorsal horn of the spinal cord. Other studies have also shown that the TLR4 was not predominantly expressed in the superficial layer [52,53]. Therefore, future studies with larger sample sizes and diverse animal models are needed to confirm the expression of TLR4 in the spinal cord dorsal horn. However, our IF staining showed that TRPV1 expression was predominantly located in the superficial dorsal horn of the spinal cord, which is consistent with previous studies [28]. Third, our methodology was based on a study by Li et al. [54] that focused on single-dose pharmacological blocking of TLR4. A dose–response approach is necessary for future studies to gain more mechanistic information. Moreover, Szabo-Pardi et al. [55] utilized a transcriptional blocker inserted into the TLR4 gene in between exon 2 and 3 to create a TLR4 null-reactivatable model and demonstrate the role of TLR4 in the early development of neuropathic pain. The use of TLR4-knockout mice can be considered to further investigate the effects of HBOT on alleviating CIPN.

## 4. Materials and Methods

### 4.1. Animals and Experimental Grouping

This animal study was approved by the Institutional Animal Care and Use Committee (IACUC) of Kaohsiung Medical University (IACUC Approval Number: 110229). On day 0 (D0), 36 male Sprague Dawley rats (180–200 g) (BioLASCO Taiwan Co., Ltd., Taipei, Taiwan) were randomly and evenly divided into six groups, the control group, PAC group, PAC/LPS group, PAC/LPS/HBOT group, PAC/TAK-242 group and PAC/TAK-242/HBOT group. The protocol used in each group is shown in Figure 1A. All rats were housed and cared for at the Animal Center of KMU and were provided ad libitum access to animal feed and water. Before the 14-day experiment began, the rats were housed under a 12 h light–dark cycle for 1 week at 22 °C for acclimation. On D14, all rats were sacrificed through anesthesia with Zoletil (50 μg/g, Virbac Laboratory, Crookwell, Australia) administered subcutaneously.

### 4.2. Chemotherapy-Induced Periphery Neuropathy (CIPN) Rat Model

For inducing the CIPN model, rats were treated with PAC (6 mg/mL, Intaxel, Fresenius Kabi Oncology, Gurugram, India) diluted with 0.9% normal saline into 1 mg/mL. Four 2 mg/kg injections were administered intraperitoneally on D0, D2, D4 and D6, respectively (Figure 1A). The control group received the same volume of 0.9% normal saline vehicle. LPS (5 mg/kg, intraperitoneal) was administered once on D0. Rats received TAK-242 (20 µg, intrathecal) through a sustained-release pump implanted intrathecally (ALZET Osmotic Pumps, Cupertino, CA, USA).

### 4.3. Hyperbaric Oxygen Therapy (HBOT)

HBOT was administered once per day from D7 to D13 (Figure 1A) in an HBOT chamber (Genmall Biotechnology, Taipei, Taiwan). After the rats were put into the chamber, the inner pressure was gradually elevated to 2.5 ATA over 20 min and then maintained for 1 h. Partition plates were inserted in the chamber to prevent rats from crowding together and to ensure that each rat could breathe freely. After 1 h of treatment, the pressure in the HBOT chamber was gradually reduced to normal room pressure over 20 min.

### 4.4. Assessment of Mechanical and Thermal Withdrawal Threshold

From D0 to D14, all rats underwent behavioral testing every 2 days (Figure 1A). To adapt the rats to the test environment, each rat was placed in the testing chamber for at least 10 min prior to the test. Mechanical allodynia was assessed using a Plantar Dynamic Aesthesiometer (Model 37370; Ugo Basile, Comerio, Italy) and the threshold time of paw withdrawal. A metal rod (2 mm diameter) was used to apply pressure on the right middle plantar surface through the metal mesh. The pressure increased at a rate of 2.5 g/s and stopped as soon as the rat retracted its paw. The limit was established at 50 g. To evaluate the thermal allodynia, the heat stimulation withdrawal threshold time was measured using a Hargreaves apparatus (Model 37370; Ugo Basile, Comerio, Italy), which used an infrared laser beam directed at 190 mW/cm^2^ to heat a glass plate on which the right paw was placed. The cut-off time was 20 s to avoid burn injury. Each measurement was taken before drug injection and HBOT, and was repeated at least five times at 10 min intervals, with at least 20 min of rest before the next hind paw application.

### 4.5. Western Blots Assay

Western blots of spinal cord dorsal horn sections with TLR4 (1: 1000, Proteintech, Rosemont, IL, USA, 19188-1-AP), MyD88 (1:1000 Proteintech, 23230-1-AP), p-NF-κB (1:1000, cell signaling, 3033), NF-κB (1:1000, cell signaling, 8482) and TRPV1 (1: 1000; Novus Biologicals, Centennial, CO, USA, NBP1-97417) were produced. The protein bands in the blot were visualized using the ECL Western Blotting Detection Kit and Bio-Rad ChemiDoc XRS system. The band intensity was quantified and plotted using Quantity One Software (Version 4.6.8, Bio-Rad, Hercules, CA, USA).

### 4.6. IF Assay

On D14 (Figure 1A), tissues from the L3–L5 segments of the spinal cord and DRGs were collected and fixed in 4% paraformaldehyde overnight. Before their transfer to a 30% sucrose solution for 24 h, the tissues were washed with 1× phosphate-buffered saline (PBS, diluted from 10× PBS, Biomate, BR110-1L, Taipei, Taiwan, R.O.C.) at 4 °C. The solution was changed twice, and the tissues were frozen. Cryostat samples were cut into 20 μm transverse sections for IF staining. The spinal cord sections were blocked in 5% normal goat serum, 2% bovine serum albumin and 0.1% Triton X-100 for 1 h at room temperature. They were incubated with the following primary antibodies at 4 °C overnight: TLR4 (1:200, Proteintech, 19188-1-AP), TRPV1 (1:200, Novus Biologicals, NBP1-97417) and MYD88 (1:200, Proteintech, 23230-1-AP). After 1 day, the secondary antibodies were added as described in a previous report [56]. To detect TLR4 and TRPV1 in the DRGs, the tissue sections were incubated with a combination of anti-NeuN (1:500, Merck Millipore, Burlington, MA, USA, MAB377), TLR4 and TRPV1 overnight at 4 °C. The sections of the sciatic nerve were incubated with MBP (1:250, Merck Millipore, Ab5864) and NF200 (1:500, Sigma, St. Louis, MO, USA, N5389). After one night, the primary antibodies were washed off with 1× PBS 3 times, and the secondary antibodies were added as described in a previous report [56]. Subsequently, the mixture was maintained at room temperature for 1 h. After this, the excess of secondary antibodies was washed off with 1× PBS 3 times. Images were captured applying a fluorescence microscope. Image Pro Plus 6.0 (IPP 6.0) was used for calculating the percentage of fluorescent cells in the spinal cord dorsal horn.

### 4.7. Statistical Analysis

Statistical analysis was performed using SPSS (ver. 14.0, USA, RRID: SCR 019096). The experimental results are presented as mean and standard error of the mean (SEM). The mean values of each group are displayed using bar graphs. Student’s *t* test was applied for analyzing the results of IF staining and the behavior test. A statistically significant difference was defined as a *p* value of 0.05.

## 5. Conclusions

Our results demonstrated the antinociceptive effect of HBOT on mechanical and heat hyperalgesia in the PAC- and PAC/LPS-treated rats. This study suggested that HBOT can ameliorate PAC-related CIPN by downregulating TLR4/MyD88/NF-κB signaling and TRPV1 activation in the CIPN rat model.

## Figures and Tables

**Figure 1 ijms-24-05379-f001:**
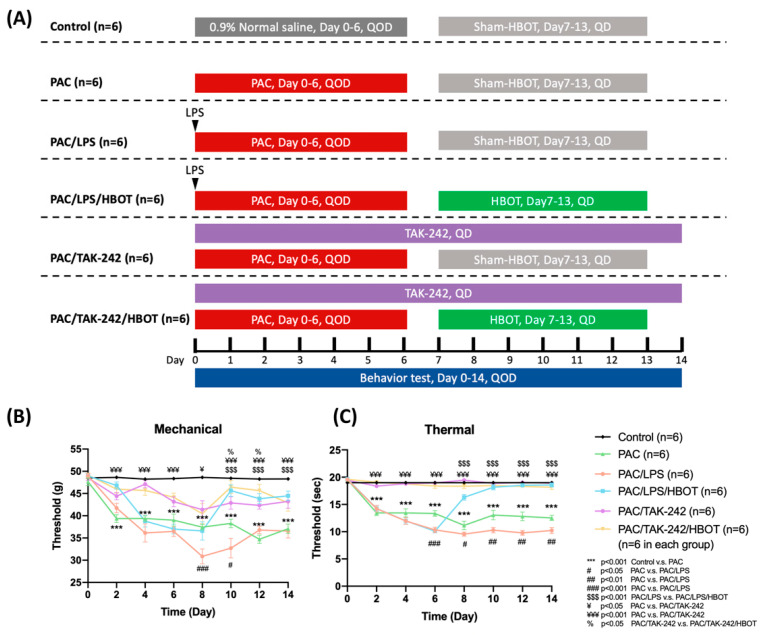
PAC-induced mechanical allodynia and thermal hyperalgesia as determined using the von Frey test and paw thermal stimulation test, respectively. HBOT improved mechanical allodynia and thermal hyperalgesia evoked by PAC. (**A**) Experimental grouping and protocol for the establishment of the rat model of PAC-induced neuropathic pain as well as LPS, TAK-242 and HBOT treatment. (**B**) Effect of LPS, TAK-242 and HBOT on mechanical allodynia in PAC-treated rats. (**C**) Effect of LPS, TAK-242 and HBOT on thermal hyperalgesia in PAC-treated rats. Data are expressed as mean ± SEM (*n* = 6 per group). PAC: Paclitaxel, SEM: standard error of the mean.

**Figure 2 ijms-24-05379-f002:**
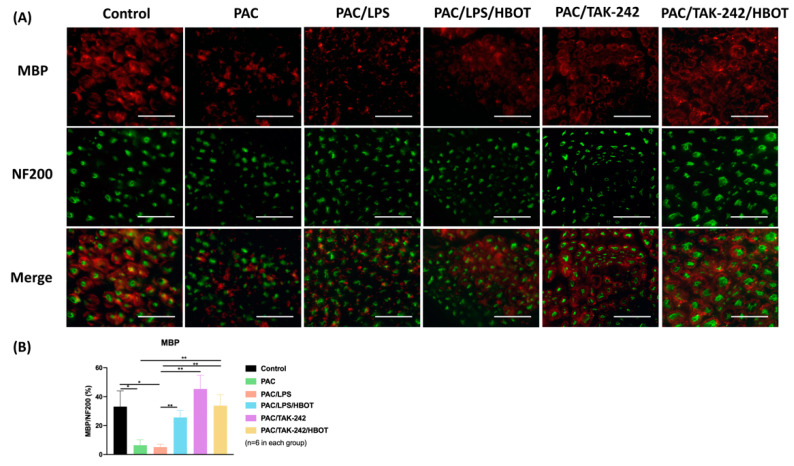
Expression of myelin basic protein (MBP) in rat sciatic nerve. (**A**) Immunofluorescence (IF) images of MBP and NF-200 in each experimental group. The IF staining of MBP (a marker of the myelin sheath, indicated by red) and NF-200 (a marker of large-diameter myelinated nerve fibers, indicated by green) were performed. Scale bar = 100 μm. (**B**) Representative bar graph illustrating the MBP/NF-200 ratio of fluorescence intensity. Data are expressed as mean ± SEM (*n* = 6 per group). * *p* < 0.05, ** *p* < 0.01. MBP: myelin basic protein. NF200: neurofilament 200. SEM: standard error of the mean.

**Figure 3 ijms-24-05379-f003:**
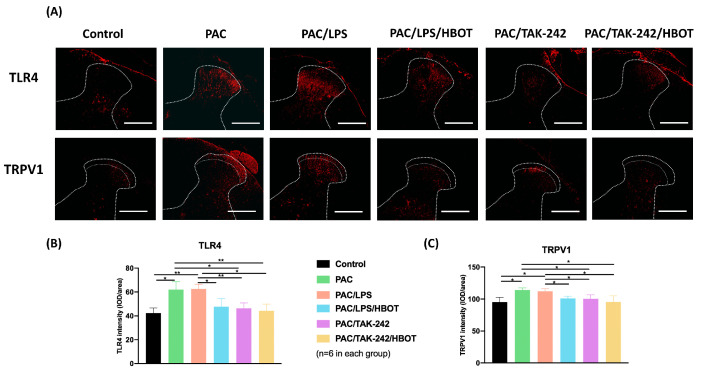
Expression of TLR4 and TRPV1 in the rat spinal cord dorsal horn. (**A**) IF image indicating TLR4 and TRPV1 antibody staining of the L3–5 spinal cord from the control, PAC, PAC/LPS, PAC/LPS/HBOT, PAC/TAK-242 and PAC/TAK-242/HBOT groups. Scale bar = 200 μm. (**B**) Bar graphs showing the intensity of TLR4 antibody staining. (**C**) Bar graphs showing the intensity of TRPV1 antibody staining. Data are expressed as mean ± standard error of the mean (SEM) (*n* = 6 per group). * *p* < 0.05, ** *p* < 0.01.

**Figure 4 ijms-24-05379-f004:**
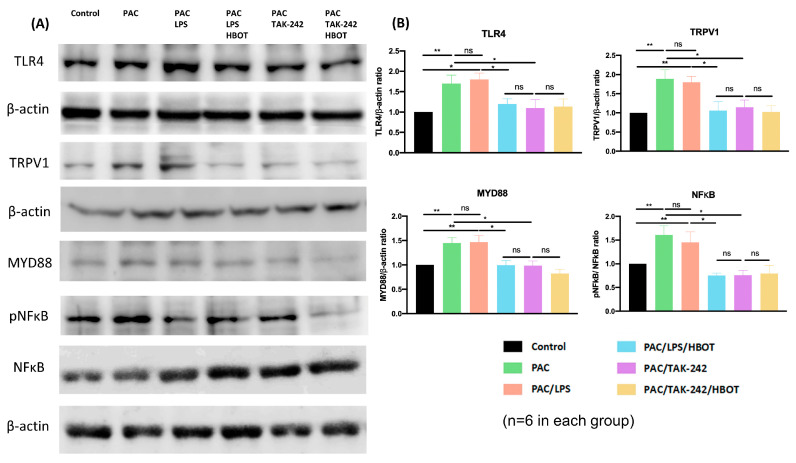
The expression of TLR4, TRPV1, MyD88, pNF-κB and NF-κB in the rat spinal cord dorsal horn. (**A**) Western blotting analysis of TLR4, TRPV1, MyD88, pNF-κB and NF-κB. (**B**) Representative bar graphs illustrating the quantitative analysis of Western blotting results. Significantly increased expression of TLR4, TRPV1, MyD88 and pNF-κB/NF-κB in the PAC and PAC/LPS groups compared to other groups. Each bar represents mean ± SEM (*n* = 6 per group). * *p* < 0.05, ** *p* < 0.01, ns: not significant.

**Figure 5 ijms-24-05379-f005:**
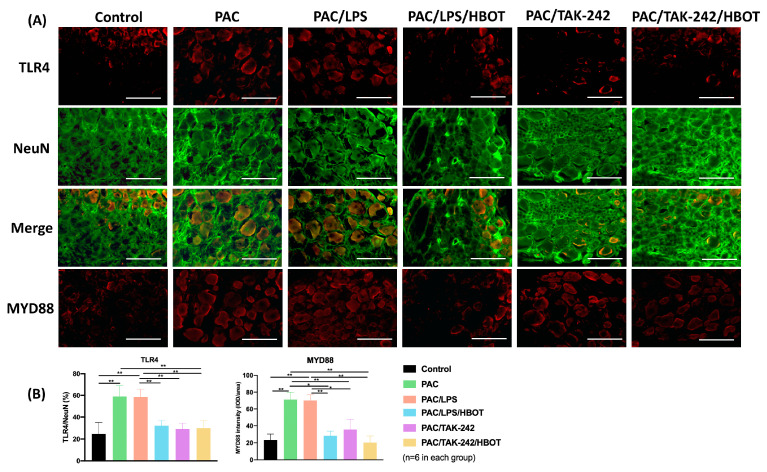
Expression of TLR4 and MyD88 in the rat DRG. (**A**) IF images indicating TLR4 and MyD88 antibody staining from the control, PAC, PAC/LPS, PAC/LPS/HBOT PAC/TAK-242, and PAC/TAK-242/HBOT groups. IF staining of TLR4 (indicated by red) and DRG neurons (NeuN, indicated by green) was performed separately and then merged. Scale bar indicates 100 μm. (**B**) Representative bar graphs illustrating the percentage of TLR4 and MyD88. Error bars, mean ± SEM (*n* = 6 per group). * *p* < 0.05, ** *p* < 0.01 vs. control group. IF: immunofluorescence. DRG: dorsal root ganglion.

**Figure 6 ijms-24-05379-f006:**
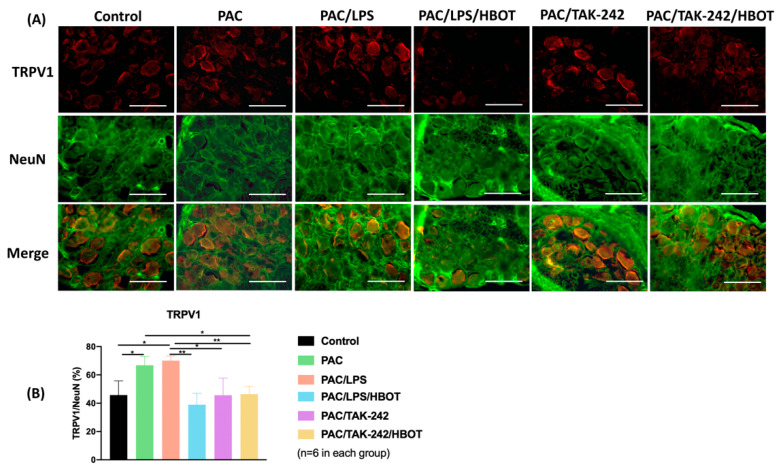
TRPV1 expression in rat DRG. (**A**) IF images indicating TRPV1 antibody staining from the control, PAC, PAC/LPS, PAC/LPS/HBOT, PAC/TAK-242 and PAC/TAK-242/HBOT groups. IF staining of TRPV1 (indicated by red) and DRG neurons (NeuN, indicated by green) were performed. Scale bar indicates 100 μm. (**B**) Representative bar graphs illustrating the percentage of TRPV1. Error bars, mean ± SEM (*n* = 6 per group). * *p* < 0.05, ** *p* < 0.01 vs. control group. IF: immunofluorescence. DRG: dorsal root ganglion.

**Figure 7 ijms-24-05379-f007:**
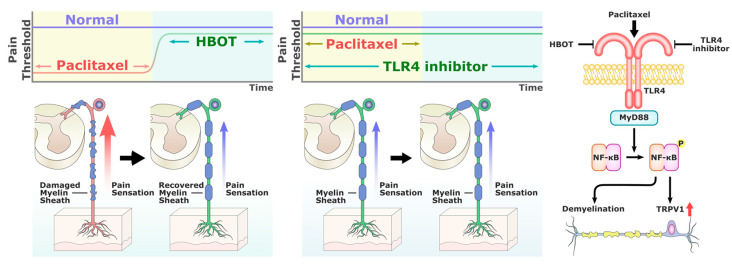
Summary of the effectiveness of HBOT in alleviating PAC-induced neuropathy by suppressing TLR4/MyD88/NF-κB signaling and TRPV1 activation, and reducing axon demyelination.

## Data Availability

The data presented in this study are available upon request from the corresponding authors.

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
