# Peer review of "Hyperbaric Oxygen Therapy Alleviates Paclitaxel-Induced Peripheral Neuropathy Involving Suppressing TLR4-MyD88-NF-κB Signaling Pathway"

_ijms, 2023, doi:10.3390/ijms24065379_

Round 1

Reviewer 1 Report (Previous Reviewer 2)

The manuscript demonstrates that hyperbaric therapy and TLR4 antagonist (TAK-242) ameliorated the paclitaxel-induced long-term chemotherapy-induced peripheral neuropathy, via downregulation of TLR4 and TRPV1. The data presented evidence also that HBOT ameliorates chemotherapy-induced peripheral neuropathy by suppressing the TLR4-MyD88-NF-κB axis. Preventing or limiting chemotherapy-induced peripheral neuropathy is an urgent need and studies aimed at this goal are valuable

In my opinion, the manuscript has been significantly improved.

Minor remarks:

Line 75: Is “that” needed?

Line 283: “as the methods designed”, not clear, please improve the sentence

What is 0.1 M PBS? If you mean phosphate-buffered saline (the acronym should be explained), it is not a single compound so molarity is not appropriate; better, the composition of PBS could be reported somewhere.

Line 338: Please change “um” to “μm”

Line 348: Was the excess of secondary antibodies washed off?

Lines 350/351: “for calculating and determining the percentage of fluorescent cells”, what is the difference between calculating and determining?

Author Response

Thank you very much for your comments and suggestions on our manuscript. Please see the attachment.

Reviewer 2 Report (New Reviewer)

This is an interesting and well conducted study, with limitations as noted in the Discussion section (+ some other). 

Poor quality of the included graphs.  I presume the quality will be considerably improved in the final version.  

The mechanistic conclusion, based on single dose experiments, which for me as a traditional pharmacologist weakens the conclusion, where the basic premise of pharmacology is replaced with "all-or-none" responses. When it comes the used rat model of PAC-induced peripheral neuropathy, the authors refer to Li et al (reference 53). However, this study is  still based on single doses. A dose response-approach should most probably give essential mechanistic information! 

Without being an HBOT expert, I am wondering why this treatment is not like "pouring gasoline at the fire"? Peripheral neuropathy results from an inflammatory process, characterized by a high degree of oxidative and nitrosative stress, creating a situation where it is far from given that HBOT will offer any improvement.

As far as I understand, PAC causes peripheral neuropathy by stabilizing microtubles in DRG neurons in such a way that axonal transport is disrupted.  Could HBOT reverse neuropathy by microtuble depolymerization, through a pure hyperbaric mechanism?

Nevertheless, the present manuscript represents a  good piece of scientific work! 

Author Response

Thank you very much for your comments and suggestions on our manuscript. Please see the attachment.

This manuscript is a resubmission of an earlier submission. The following is a list of the peer review reports and author responses from that submission.

Round 1

Reviewer 1 Report

Major

These results of the manuscript were not enough evidence to support the hypothesis “Hyperbaric Oxygen Therapy Alleviates Paclitaxel-Induced Peripheral Neuropathy Involving Suppressing TLR4-MyD88-NF-κB Signaling Pathway”. The purpose of this study focus on pain or inflammation should be demonstrated more clearly. For example, the sentence “….for confirming the anti-inflammatory effect of HBOT via modulation of TLR4/MyD88/NF-κB and TRPV1 pathway” (Page 7, Line 190-193) was ambiguous. Moreover, no synergistic effect of TAK-242 and HBOT was observed in this study, why?

Minor

1. In the Method section (Page 10, Paragraph 1), how to calculate the positive cells should be demonstrated in detail.

2. All figures should be demonstrated the number of animals.

3. In Figure 3, the TLR4 positive cells has been stained in different laminae of SCDH, the histogram of TLR4 positive intensity in the superficial and deep laminae should be shown respectively. Moreover, the reason should be discussed in the discussion section.

4. In Figure 4, better representative WB bands should be shown and “b-actin” in the Y-axis should be rectified.

5. In Figure 5 and 6, the NeuN staining should be located in the cytoplasm and nucleus of DRG neuron. The IHC figures were seemed to be false positive…

Author Response

Thank you for the detailed comments and suggestions.
Please see the attachment.

Reviewer 2 Report

The manuscript demonstrates that hyperbaric therapy and TLR4 antagonist (TAK-242) ameliorated the paclitaxel-induced long-term chemotherapy-induced peripheral neuropathy, via downregulation of TLR4 and TRPV1. The data presented evidence also that HBOT ameliorates chemotherapy-induced peripheral neuropathy by suppressing the TLR4-MyD88-NF-κB axis. Preventing or limiting chemotherapy-induced peripheral neuropathy is an urgent need and studies aimed at this goal are valuable. I am even not sure if IJMS is optimal for publishing these results and if a medical journal devoted to cancer/chemotherapy would not be a better place. However, the manuscript fits the scope of the journal (field of molecular medicine).

Remarks:

The authors refer to the their previous paper (Ref. 18) in the methodical part. It is legitimate but as this source is not freely available I suggest brief descriptions of the so referred methods in the manuscript.

Figures 1 and 4. Statistically significant differences with respect to Control are indicated. It would be desirable to indicate also significance of differences between PAC and various treatments.

Line 197: “can significantly ameliorated”

Line 272: “Rats were housed under a 12-h light-dark cycle for 1 week at 22°C”. It is somewhat unclear: Was it before the 14-day experiment?

Author Response

Thank you for the detailed comments and advice.

Round 2

Reviewer 1 Report

These results of the manuscript were not enough evidences to support the hypothesis and draw the conclusion. Please resubmit the manscript after the supplemention of additional experimental evidences.